# Pitfalls in the Application of Dispase-Based Keratinocyte Dissociation Assay for In Vitro Analysis of Pemphigus Vulgaris

**DOI:** 10.3390/vaccines10020208

**Published:** 2022-01-28

**Authors:** Morna F. Schmidt, Maria Feoktistova, Diana Panayotova-Dimitrova, Ramona A. Eichkorn, Amir S. Yazdi

**Affiliations:** 1Department of Dermatology and Allergology, University Hospital RWTH Aachen, 52074 Aachen, Germany; morschmidt@ukaachen.de (M.F.S.); mfeoktistova@ukaachen.de (M.F.); ddimitrova@ukaachen.de (D.P.-D.); 2Department of Dermatology, University Hospital Tuebingen, 72076 Tuebingen, Germany; ramona.stach@med.uni-tuebingen.de

**Keywords:** keratinocytes, desmosomes, autoimmune bullous diseases, pemphigus, in vitro analysis

## Abstract

Pemphigus vulgaris (PV) is a chronic, life-altering autoimmune disease due to the production of anti-desmoglein antibodies causing the loss of cell–cell adhesion in keratinocytes (acantholysis) and blister formation in both skin and mucous membranes. The dispase-based keratinocyte dissociation assay (DDA) is the method of choice to examine the pathogenic effect of antibodies and additional co-stimuli on cell adhesion in vitro. Despite its widespread use, there is a high variability of experimental conditions, leading to inconsistent results. In this paper, we identify and discuss pitfalls in the application of DDA, including generation of a monolayer with optimized density, appropriate culturing conditions to obtain said monolayer, application of mechanical stress in a standardized manner, and performing consistent data processing. Importantly, we describe a detailed protocol for a successful and reliable DDA and the respective ideal conditions for three different types of human keratinocytes: (1) primary keratinocytes, (2) the HaCaT spontaneously immortalized keratinocyte cell line, and (3) the recently characterized HaSKpw spontaneously immortalized keratinocyte cell line. Our study provides detailed protocols which guarantee intra- and inter-experimental comparability of DDA.

## 1. Introduction

Pemphigus vulgaris (PV) is a chronic, bullous autoimmune dermatosis with dysregulated skin integrity due to autoantibody-induced loss of cell adhesion, called acantholysis [1]. Since the disease impairs the epidermal barrier function with a risk of superinfection and massive weight loss, it can be fatal without therapy [2]. Desmosomes are complex protein structures composed of desmosomal cadherines (desmoglein, desmocollin), armadillo family proteins (plakoglobin, plakophilin), and desmoplakin, which provides the link to keratin filaments [3,4]. The skin is exposed to high mechanical stress and serves as the first line of defense against environmental factors, and therefore proper desmosome function is of critical importance for skin integrity [5]. In concert with tight junctions and adherent junctions, desmosomes ensure skin resistance to physical stress and govern epithelial differentiation and function [3,4,6,7].

The hallmark of PV is the production of autoantibodies against calcium-dependent desmosomal cadherines: desmogleins 3 (Dsg3) and 1 (Dsg1) [4,8]. Dsg3 is largely expressed in supra-basal epidermal skin and mucous membranes, while Dsg1 is predominantly present in sub-corneal layers of the skin epidermis. Depending on the expression pattern of autoantibodies, different clinical types and histologically correlated features are observed in pemphigus patients [4,9]. In PV, secretion of anti-Dsg1 and -Dsg3 antibodies leads to supra-basal acantholysis with clinically associated flaccid blisters and secondary erosions. It was demonstrated that the pathogenesis of pemphigus could result from a direct inhibition of desmoglein through the antibody-binding to Dsg1 and Dsg3, with resulting impaired desmosomal trans-interaction [10,11]. Furthermore, both anti-Dsg1 and -Dsg3 antibodies are involved in the activation of additional signaling pathways, resulting in a modulation of keratinocyte cell adhesion properties [4,12,13]. Despite these advances in our understanding, the exact mechanism of antibody-mediated acantholysis remains incomplete, making a reliable in vitro testing platform indispensable.

The dispase-based keratinocyte dissociation assay (DDA) is currently the main tool for analysis of antibody-induced acantholysis in PV in vitro. The assay was first reported by Calautti et al. in 1998 in the context of cell–cell contact studies [14]. In 2001, DDA was first applied in pemphigus research [15]. The principle of this method is to investigate the influence of different stimuli on the loss of keratinocyte adhesion in cell monolayers. DDA includes five main steps: (1) generation of a cell monolayer in certain culturing conditions, (2) incubation of the cell monolayer with various stimuli, such as anti-Dsg3 antibody, (3) detaching of cell monolayers from the bottom of the cell culture dish, (4) application of mechanical stress by standardized pipetting for monolayer fragmentation, and (5) quantification of fragments and data processing. Depending on the respective study, primary or immortalized keratinocyte cell lines can be used as a model system for DDA. Although DDA is mostly used for the analysis of cell adhesion in vitro, there is a wide range of different applications [14,15,16,17,18]. Typical challenges when performing a DDA are as follows: generation of a monolayer with optimal density, appropriate culturing conditions to prevent the generation of an unbreakable or too-fragile cell monolayer, application of mechanical stress in a standardized manner, and finally, consistent data processing.

Here, we present detailed protocols for performing a successful DDA and compare optimal conditions for primary keratinocytes (PK), HaCaT, a spontaneously immortalized keratinocyte cell line, and the recently characterized HaSKpw spontaneously immortalized keratinocyte cell line [19]. Our experiments demonstrate that each individual step must be adapted to the respective cell line and its characteristic properties. We aim to standardize the experimental protocols to ensure intra- and inter-experimental comparability.

## 2. Materials and Methods

### 2.1. Cell Lines and Antibodies

HaCaT spontaneously immortalized human keratinocyte cell line (kindly provided by Prof. P. Boukamp, formerly DKFZ Heidelberg)HaSKpw spontaneously immortalized human keratinocyte cell line (kindly provided by Prof. P. Boukamp, formerly DKFZ Heidelberg)Primary human keratinocytes (ethical approval: 547/2011BO2)AK23 anti-Dsg3 antibody (hybridoma cell supernatants were kindly provided by Prof. R. Tikkanen, Justus Liebig University Giessen, Germany, and purification was performed following Beckert et al. [20])Human immunoglobulin isotype control (Thermo Fisher Scientific Inc., Waltham, MA, USA, Cat. no. 02-7102)

### 2.2. Reagents and Equipment

24-Well Cell Culture Multiwell Plate(sterile, with lid) (Greiner AG, Kremsmünster, Austria; Cat. no. 10177380)Antibiotics/antimycotics (100×) (Thermo Fisher Scientific Inc., Waltham, MA, USA, Cat. No. 15240062)Calcium chloride (CaCl_2_) (Carl Roth, Karlsruhe, Germany, Cat. No. CN93.1)Chelex 100 Chelating Resin, analytical grade, 100–200 mesh, sodium form (Bio-Rad Laboratories Inc., Hercules, CA, USA, Cat. No. 1422832)ChemiDoc Imaging System (Bio-Rad Laboratories Inc., Hercules, CA, USA, Cat. No. 1708195)CnT-Prime Epithelial Proliferation Medium (CELLnTEC ADVANCED CELL SYSTEMS AG, Bern, Switzerland, Cat. No. CnT-PR)Dispase II powder (Thermo Fisher Scientific Inc., Waltham, MA, USA, Cat. No. 17105041)Dulbecco’s Modified Eagle Medium (DMEM) (PAN-Biotech GmbH, Aidenbach, Germany, Cat. No. P04-04515)Eppendorf Xplorer^®^ electrical pipette (Eppendorf AG, Hamburg, Germany, Cat. No. 4861000040)Fetal Bovine Serum (FBS) standard (PAN-Biotech GmbH, Aidenbach, Germany, Cat. No. P30-3306), is equivalent toFetal Calf Serum (FCS)Hanks’ Balanced Salt Solution (HBSS) with Ca^2+^ (Thermo Fisher Scientific Inc., Waltham, MA, USA, Cat. No. 14025-050)Syringe filters, ROTILABO^®^ PVDF, 0.22 µm (Carl Roth, Karlsruhe, Germany, Cat. No. P666.1)Thiazolyl blue tetrazolium bromide (MTT) (Invitrogen by Thermo Fisher Scientific Inc., Waltham, MA, USA, Cat. No. M6494)Whatman^®^ Grade 1 Qualitative Filter Paper (Merck KGaA, Darmstadt, Germany, Cat. No. WHA1001325)

### 2.3. Reagents Setup

Chelated FCS: Swell 100 g of Chelex 100 Chelating Resin in 500 mL of dH_2_O. Adjust the pH to 7.4 with HCl while stirring (pH will take a while to stabilize during titration). Filter through Whatman^®^ Grade 1 Qualitative Filter Paper. Scrape resin slurry into 500 mL of FCS and stir at room temperature for 3 h or at 4 °C overnight. Filter the chelated FCS through Whatman^®^ Grade 1 Qualitative Filter Paper and discard the resin slurry. Filter the chelated FCS through a 0.22 µm filter to sterilize it. Aliquot sterile chelated FCS and store it at −20 °C.CnT-Prime Epithelial Proliferation Medium supplemented with 1% antibiotics/antimycotics: 5 mL of antibiotics/antimycotics per 500 mL bottle of medium.Dispase solution: dissolve the dispase in HBSS buffer to a final concentration of 2.5 U/mL.DMEM supplemented with 10% FCS: 50 mL of FCS per 500 mL bottle of medium.DMEM supplemented with 8% chelated FCS: 40 mL of chelated FCS per 500 mL bottle of medium.MTT staining solution: a 5 mg/mL working solution is prepared by dissolving MTT in HBSS buffer.

### 2.4. Software

ImageJ (Rasband, W.S., ImageJ, US National Institutes of Health, Bethesda, MD, USA, https://imagej.nih.gov/ij/, 1997–2021; accessed on 6 December 2021)Image Lab Software (Bio-Rad Laboratories Inc., Hercules, CA, USA, Cat. No. 1708195)Microsoft Excel (Microsoft Corporation, Redmond, Washington, DC, USA, version 2016)

### 2.5. Methods and Protocols

#### 2.5.1. Cell Culture

Primary human keratinocytes were isolated and pooled from different donors to minimize individual-specific influencing factors and to achieve an average outcome with reproducible experimental conditions. Experiments should be performed between passages 3 and 8 [21]. Cells were cultured in CnT-Prime Epithelial Proliferation Medium without FCS supplemented with antibiotics/antimycotics at 37 °C in 5% CO_2_ atmosphere.

HaCaT and HaSKpw spontaneously immortalized human keratinocyte cell lines were cultured in DMEM supplemented with 10% FCS at 37 °C in 5% CO_2_ atmosphere.

#### 2.5.2. Dispase-Based Keratinocyte Dissociation Assay in Primary Keratinocytes

##### Generation of a Cell Monolayer, Time Consideration: 2–3 Days

Note: ensure sterile working environment.

Dilute 6 × 10^5^ cells/well in keratinocyte medium (CnT-Prime Epithelial Proliferation Medium) without FCS supplemented with antibiotics/antimycotics. Plate 1 mL of cell suspension per well in a sterile 24-well plate.Incubate the cells at 37 °C in 5% CO_2_ atmosphere. The cells are usually confluent by 24–48 h. Medium should be changed every day.When the cells reach confluency, aspirate the medium and replace with a new medium (CnT-Prime Epithelial Proliferation Medium) containing 1.8 mM of CaCl_2_.Incubate the cells at 37 °C in 5% CO_2_ atmosphere for an additional 24 h.

##### Stimulation of the Monolayer, Time Consideration: Depending on the Stimulation Procedure

Note: ensure sterile working environment.

Aspirate the medium and add the stimulus diluted in medium (CnT-Prime Epithelial Proliferation Medium) without supplemented CaCl_2_.

Note: in this experimental setting, monolayers were stimulated with either human IgG human isotype control (IgG), 30 µg/mL, or AK23 anti-Dsg3 antibody, 30 µg/mL, at 37 °C in 5% CO_2_ atmosphere for 4 h.

Note: add at least 250 µL of medium/well to fully cover the bottom.

##### Detaching of the Monolayer from the Cell Culture Dish, Time Consideration: Approximately 50 min

Aspirate the medium and rinse the cell layer with 500 µL of HBSS.Add 350 µL of dispase solution to each well.Incubate the plate for 30–40 min at 37 °C in 5% CO_2_ atmosphere. The monolayer should be completely detached from the dish bottom.

Application of Mechanical Stress, Time Consideration: Approximately 30 min.

Aspirate dispase solution with a pipette and add 350 µL of HBSS into each well.Add 20 µL of MTT staining solution per well and incubate for 10–15 min at 37 °C in 5% CO_2_ atmosphere.Remove the staining solution and add 350 µL of HBSS per well.Apply mechanical stress by pipetting with an electrical pipette (settings: Pip, speed 8, 300 µL).

Note: pipette a clean pipette tip in HBSS buffer to wet the tip to assure that no fragments will stick to the tip wall.

Quantification of Fragments and Data Processing (See Section 2.5.4).

#### 2.5.3. Dispase-Based Keratinocyte Dissociation Assay in HaCaT and HaSKpw Keratinocytes

##### Generation of a Cell Monolayer, Time Consideration: 1 Day (HaCaT) or 2–3 Days (HaSKpw)

Note: ensure sterile working environment.

Dilute 6 × 10^5^ cells/well in DMEM supplemented with:-Chelated FCS (1.6 mM final concentration of Ca^2+^) for HaCaT cell culture-Standard FCS (1.9 mM final concentration of Ca^2+^) for HaSKpw cell culture

Plate 1 mL of cell suspension per well, in a 24-well plate. The cells are usually confluent by 24–48 h. HaSKpw cells should be further cultivated for 24–48 h to increase layer stability. Change medium every day.

Note: the appropriate calcium concentration and the culturing time are of critical importance.

##### Stimulation of the Monolayer, Time Consideration: Depending on the Stimulation Procedure

Note: ensure sterile working environment.

Aspirate the medium and add the stimulus diluted in the respective medium:-Chelated FCS (1.6 mM final concentration of Ca^2+^) for HaCaT cells-Standard FCS (1.9 mM final concentration of Ca^2+^) for HaSKpw cells

Note: in this experimental setting, monolayers were incubated with:

-Either human IgG human isotype control (IgG), 20 µg/mL, or AK23 anti-Dsg3 antibody, 20 µg/mL, at 37 °C in 5% CO_2_ atmosphere for 4 h for HaCaT cells-Either human IgG human isotype control (IgG), 30 µg/mL, or AK23 anti-Dsg3 antibody, 30 µg/mL, at 37 °C in 5% CO_2_ atmosphere for 4 h for HaSKpw cells

Note: add at least 250 µL of medium/well to fully cover the bottom.

##### Detaching of the Monolayer from the Cell Culture Dish, Time Consideration: Approximately 50 min

Aspirate the medium and rinse the cell layer with 500 µL of HBSS.Add 350 µL of dispase solution to each well.Incubate the plate at 37 °C in 5% CO_2_ atmosphere for:-Approximately 40 min (HaCaT cells)-Approximately 30 min (HaSKpw cells)

After the respective incubation time, the monolayer should be completely detached from the dish bottom.

##### Application of Mechanical Stress, Time Consideration: Approximately 30 min

Aspirate dispase solution carefully with a pipette and apply 350 µL of HBSS in each well.Add 20 µL of MTT staining solution per well and incubate for 10–15 min at 37 °C in 5% CO_2_ atmosphere:-Approximately 10 min for HaCaT monolayer-Approximately 20 min for HaSKpw monolayerCarefully remove the staining solution and add 350 µL of HBSS per well.Apply mechanical stress by pipetting with an electrical pipette (settings: Pip, speed 8, 300 µL).

Note: pipette a clean pipette tip in HBSS buffer to wet the tip to assure that no fragments will stick to the tip wall.

Quantification of Fragments and Data Processing (See Section 2.5.4).

#### 2.5.4. Application of Mechanical Stress, Quantification of Fragments and Data Processing

Set the pipetting conditions by first applying the defined mechanical stress to the replicates within the positive control condition (in this experimental setting, anti-Dsg3-treated samples were used as a control). Take the highest number of pipetting steps necessary to fragment the monolayer between the respective replicates (at least two or more fragments) and apply the same pipetting number to all samples.Take at least three pictures of each well (Chemilum Photo Chamber, Image Lab Software).

Note: carefully shake the plate between taking the pictures in order to separate fragments that have eventually adhered to each other.

3.Quantify fragments with imaging software, i.e., ImageJ software:a.Open the image with ImageJ.b.Mark the target area with the desired shape and copy.c.Create a new file and insert the copied file (Menu: File -> New (choose white background) -> Image -> insert)d.Adjust the threshold of the image by marking only the cell layer fragments (Menu: Image -> Adjust -> Threshold: adjust)e.Count the particles (Menu: Analyze -> Analyze Particles: Define the size adapted to the requirements of the individual cell lines (e.g., 5–infinity). Tick: display results, clear results, summarize).f.Data are evaluated with Microsoft Excel software.

## 3. Results

### 3.1. Conditions for Cultivation of Primary Human Keratinocytes for Dispase-Based Keratinocyte Dissociation Assay

Based on existing protocols [14,15,17], we adapted the ideal culturing conditions for the generation of stable PK monolayers for DDA. Since human PKs proliferate, but do not differentiate in the absence of calcium [22], we used calcium-free medium for initial cultivation. The plated PK reached confluency after 24 h, which was shown microscopically by the complete coverage of the cell culture dish with the respective cells (Figure 1a, left). The calcium switch in the keratinocytes’ medium is a critical step in the investigation of the cell adhesion in primary keratinocytes, as desmosome formation is calcium-dependent [23,24]. The calcium-free medium was replaced by 1.8 mM of calcium-containing medium to initiate desmosome formation and subsequent differentiation of keratinocytes [25]. The cells were then further cultivated for 24 h until a stable monolayer developed, fit for further stimulation (Figure 1a, right). We observed an increased number of intracellular granules, which are the morphological correlate of keratinocyte differentiation [26]. Next, PK monolayers were incubated with human immunoglobulin (IgG) isotype control or anti-Dsg3 antibody AK23, respectively. No microscopic differences between the monolayers were detected before or after dispase treatment (Figure 1b). Of note, edge-originating cell sheet fragments were observed macroscopically in some of the anti-Dsg3-treated monolayers before application of mechanical stress (Figure 1c, left panel, bottom). Next, mechanical stress was applied to all samples by an equal number of pipetting steps, as explained in detail below (Figure 1c). As expected, treatment with anti-Dsg3 antibody resulted in the generation of a higher number of fragments, compared to IgG-treated monolayers (Figure 1d), suggesting a Dsg3-dependent reduction of intercellular cohesive strength in PK.

### 3.2. Conditions for Cultivation of HaCaT Cells for Dispase-Based Keratinocyte Dissociation Assay

We first aimed to detect the best number of HaCaT cells and the required time for generation of a stable monolayer, fit for further treatment. Therefore, we plated different numbers of HaCaT cells in 24-well plates. We observed that 6 × 10^5^ cells/well were required for generation of a confluent monolayer demonstrated microscopically by a completely covered culture dish surface within 18 h (overnight) (Figure 2a). Furthermore, we observed that overnight culturing of the HaCaT cells was sufficient to generate a suitable monolayer.

In contrast to PK, HaCaT cells can proliferate in a higher range of calcium concentration [27]. To determine the best calcium concentration for monolayer generation, we cultivated HaCaT cells in the following conditions: (1) standard Dulbecco’s Modified Eagle Medium (DMEM) supplemented with 10 % FCS with a final calcium concentration of 1.9 mM, or (2) DMEM supplemented with calcium-free FCS. To deplete calcium from serum, we performed a chelation by exposure of FCS to a chelating resin, as described by Lichti et al. [28]. Thus, the final calcium concentration in DMEM supplemented with chelated FCS was reduced to 1.6 mM. No morphological differences between HaCaT monolayers cultivated under both conditions were observed after overnight incubation (Figure 2b). Next, the monolayers treated under both conditions were exposed to dispase. Different calcium concentrations in the medium did not influence neither the microscopic nor macroscopic appearance of the HaCaT monolayers (Figure 2b,c). The monolayers cultured under both conditions demonstrated high resistance to mechanical stress, as no fragmentation was detected after a high number of pipetting steps (50×) (Figure 2c). Importantly, after cultivation of the monolayers with anti-Dsg3 antibody, we observed calcium-dependent differences in their fragility. Monolayers cultured in DMEM containing 1.6 mM of calcium were significantly more fragile compared to those cultured in medium with the higher concentration of 1.9 mM of calcium (Figure 2d,e). These observations demonstrate that 1.9 mM of calcium in the culture medium might cause desmosomal hyper-adhesion in HaCaT cells and suggest a calcium concentration of 1.6 mM as more appropriate for generation of HaCaT monolayers for DDA.

### 3.3. Conditions for Cultivation of HaSKpw Cells for Dispase-Based Keratinocyte Dissociation Assay

Initially, we aimed to determine the best number of HaSKpw cells and the required time for generation of a stable monolayer. Similar to HaCaT cells, 6 × 10^5^ HaSKpw cells/well in a 24-well format were required for generation of a confluent monolayer demonstrated by an entirely covered culture dish surface within 18 h (overnight) (Figure 3a). Despite the confluency reached, HaSKpw monolayers needed another 24 to 48 h to reach optimal stability for the subsequent application of DDA.

Next, we tested the following conditions regarding calcium concentration required for generation of an appropriate HaSKpw cell monolayer for DDA: (1) DMEM supplemented with 10% FCS with a final calcium concentration of 1.9 mM, or (2) DMEM supplemented with 8% chelated FCS (1.6 mM of calcium). Similar to the results in HaCaT cells, the calcium concentration did not influence the structure of HaSKpw monolayers before or after dispase treatment (Figure 3b). However, we observed a significant increase in the fragility of the monolayers grown in medium with chelated FCS before treatment with anti-Dsg3 antibody (Figure 3c,d). These data suggest that a calcium concentration in DMEM medium of 1.6 mM is not sufficient to generate a stable enough HaSKpw monolayer for DDA. Therefore, we excluded this condition from our following experiments and used the HaSKpw monolayer cultivated in DMEM supplied with non-chelated FCS prior to treatment with anti-Dsg3 antibody. As expected, anti-Dsg3 treatment resulted in sensitization of the monolayers to mechanical stress (Figure 3e,f).

Of note, we observed very small particles which appeared after application of mechanical stress even in our control conditions, namely IgG-treated samples (Figure 3e, black arrows).

### 3.4. Application of Mechanical Stress, Quantification of Fragments, and Data Processing

A critical point of DDA is the application of mechanical stress, which results in fragmentation of the monolayer. The number of fragments correlates to the loss of cohesion induced by different stimuli. Since the application of standardized, identical mechanical stress is a major pitfall to successfully perform the DDA, we next aimed to optimize and standardize this procedure.

After dispase treatment, the monolayers of the respective cell lines (PK, HaCaT, or HaSKpw) were stained with Thiazolyl blue tetrazolium bromide (MTT), followed by application of mechanical stress using an electric pipette in a standardized manner, based on existing approaches [15,29] Alternative methods for application of mechanical stress via rotator/rocker were previously published [16,30,31]. Confirming Ishii et al. [17], the use of orbital rotation for application of mechanical stress is too mild to induce reproducible fragmentation of respective monolayers (data not shown). In summary, the application of a defined speed using an electrical pipette seems to be the method of choice for applying sheer stress in a reproducible manner. To compare different experimental conditions, the same amount of mechanical stress was applied in all settings within an experiment. For this purpose, we defined a positive pipetting control, which represents the condition in which the monolayer is expected to be fragmented. In this protocol, we chose anti-Dsg3-treated monolayers. The mechanical stress was applied in two or three wells with anti-Dsg3-treated monolayers until fragmentation started. The highest number of pipetting steps in the respective replicates was chosen and applied for all further samples. Photographs of the fragmented monolayers (Figure 1c, Figure 2c,d, Figure 3c,e) were used for further quantification. To increase the accuracy of the counting procedure in every experiment, we counted the fragments from three independent pictures for each well (Figure 4a). After each capture, the fragments were separated from each other by gently shaking the plate. Next, we used ImageJ software, which allows automatic counting of fragments with different shape and size based on predefined settings (Figure 4b).

Our experiments with primary keratinocytes rarely showed single edge-originated fragments after detachment of the monolayer from the cell culture dish and prior to the application of mechanical stress (Figure 1c, left panel, bottom). Since these fragments only occurred in anti-Dsg3 stimulated conditions, but not in the IgG-treated control condition, we ruled out a generally reduced quality of the monolayers. Due to their size, we excluded theses “fragments” from further quantification.

Of note, we demonstrated that monolayers generated from HaSKpw cells were less stable and more fragile in comparison to PK and HaCaT cells. Here, mechanical stress in control conditions (IgG stimulation) also resulted in very small fragments/particles (Figure 3e). Since these small particles were detected in the presence of an otherwise intact monolayer, we interpreted these as leathering of the cell edges of the monolayer. ImageJ software settings have been adjusted so that all fragments smaller than the largest non-specific fragments in control conditions performed with HaSKpw were excluded and applied to all analyzed conditions within one experiment.

A common challenge for data evaluation is the variability in the number of fragments between single experiments. We detected very high variability in the number of fragments between three independent DDAs with PK, despite that the very same conditions were used in all three experiments (Figure 5a). This variability results in high standard deviations (Figure 5b, first histogram). To avoid this obstacle, the number of fragments can be normalized to the fragment in a respective negative control condition, in this protocol the IgG-treated control sample (Figure 5b, second histogram). Alternatively, the numbers can be normalized to a respective positive (treated) sample, in this protocol the anti-Dsg3-treated sample (Figure 5b, third histogram).

## 4. Discussion

The overall aim of this study was to develop a protocol for conducting a DDA with defined conditions for primary human keratinocytes and two keratinocyte cell lines. Here, we addressed in detail the characteristics of each cell line to introduce the most suitable and reliable experimental setting to investigate cell adhesion in an epidermal 2D model.

PK are the most natural and physiological cells used in our assay as they are not immortalized [32]. Therefore, they should be preferentially used in DDA. Primary keratinocytes are particularly advantageous, as they are isolated directly from healthy skin and retain the morphological and the functional properties of the original tissue. The use of pooled donors can increase the comparability and reproducibility of experiments. However, common limitations of using PK as an experimental model are the tenuous handling conditions related to the increased sensitivity to changes in nutrient and growth factors, as well as short lifetime, and the incapability for sub-culturing [33].

The obstacles of PK culturing can be overcome by usage of immortalized keratinocytes, such as HaCaT and HaSKpw cell lines. The main advantages of immortalized keratinocytes are the cost effectiveness, possibility to generate reproducible results, easier handling, and unlimited growth [34]. The spontaneously immortalized keratinocyte cell line HaCaT is widely used for in vitro keratinocytes studies and is characterized by a UV-specific mutation in the tumor suppressor gene p53 (p53) [35]. HaSKpw is a recently characterized spontaneously immortalized keratinocyte cell line which in contrast to HaCaT cells, expresses wild-type p53 and has a stable haploid genotype [19]. Contrary to HaCaT cells which express high levels of desmogleins 2, 3, and 4, HaSKpw demonstrates lower expression levels, with Dsg3 as the highest expressed desmoglein [19].

The change of the extracellular calcium levels results in rapid organization of desmosomes and adherent junctions [36,37] and is a major determinant of the differentiation state of mammalian keratinocytes in vitro [38]. An appropriate calcium concentration in the cultivation medium is of critical importance for the successful application of DDA. Keratinocytes are exposed to a physiological calcium gradient under normal conditions, resulting in their division, differentiation, and ascent from stratum basale to the epidermal surface [39]. These physiological conditions are mimicked in in vitro DDA by the calcium switch from a low to high calcium concentration. In our study, we confirmed that to obtain suitable PK monolayers for performing a DDA, the calcium switch from calcium-free to a 1.8 mM-containing cultivation medium should be performed after reaching full confluency of the cell layers.

In contrast to PK, spontaneously immortalized cell lines can proliferate in a higher range of calcium concentrations and simultaneously differentiate when confluency is reached [40]. While HaSKpw cells require a medium supplemented with 1.9 mM of calcium and a prolonged cultivation time after reaching confluency, HaCaT cells display a different sensitivity for calcium. This is in line with the characteristic slower growth and lower density of HaSKpw cells compared to HaCaT cells [19]. In accordance with these data, we demonstrated that in contrast to HaSKpw cells, HaCaT cells might enter a hyper-adhesive state when cultured in 1.9 mM of calcium. This results in the generation of a hyper-stable monolayer which is not appropriate for subsequent analyses in DDA. These findings correlate with previously published reports demonstrating a hyper-adhesive state of HaCaT cells in high calcium concentrations [41,42,43].

The correct counting and quantification of the monolayer fragments resulting from the respective treatment and application of mechanical stress are also further key determinants of a reliable DDA.

A critical step during quantification of the monolayer fragments is considering the impact of the cell-specific characteristics of the respective cell line, which can influence the quality of the monolayers. For example, in monolayers generated from PK, we observed the appearance of edge-originated cell sheet fragments after detachment of the monolayer from the cell culture dish and before the application of mechanical stress. The presence of such fragments in both test and control conditions might be an indication that the monolayer has not reached the quality criteria for performing the subsequent steps of DDA. In this situation, the respective samples/wells can be excluded from further analyses. Alternatively, if fragments appear before the application of mechanical stress, but only in the test-treated samples, these very small particles can be either included or excluded in the subsequent quantification, depending on the respective research question.

Moreover, in monolayers generated with HaSKpw cells, we observed edge-originated, very small particles separated from the intact monolayer, which appeared after the application of mechanical stress even in control conditions. These particles can be acknowledged as “non-specific” and completely excluded from further quantification by application of a size cut-off to all analyzed wells. Alternatively, these small fragments can be included in the final quantification and presentation of results with respect to their size.

Altogether, the cell-specific properties and the respective scientific question must be taken into account to choose the most appropriate experimental settings.

In summary, we have provided a detailed procedure to automatically detect, count, and evaluate the resulting fragments with imaging software.

## 5. Conclusions

In this study, we provided the first detailed protocols to minimize the common inconsistencies between experiments and ensure intra- and inter-experimental comparability of DDA. In this way, desmosomal functions can be investigated to gain insights into the pathophysiology of autoimmune diseases such as PV.

## Figures and Tables

**Figure 1 vaccines-10-00208-f001:**
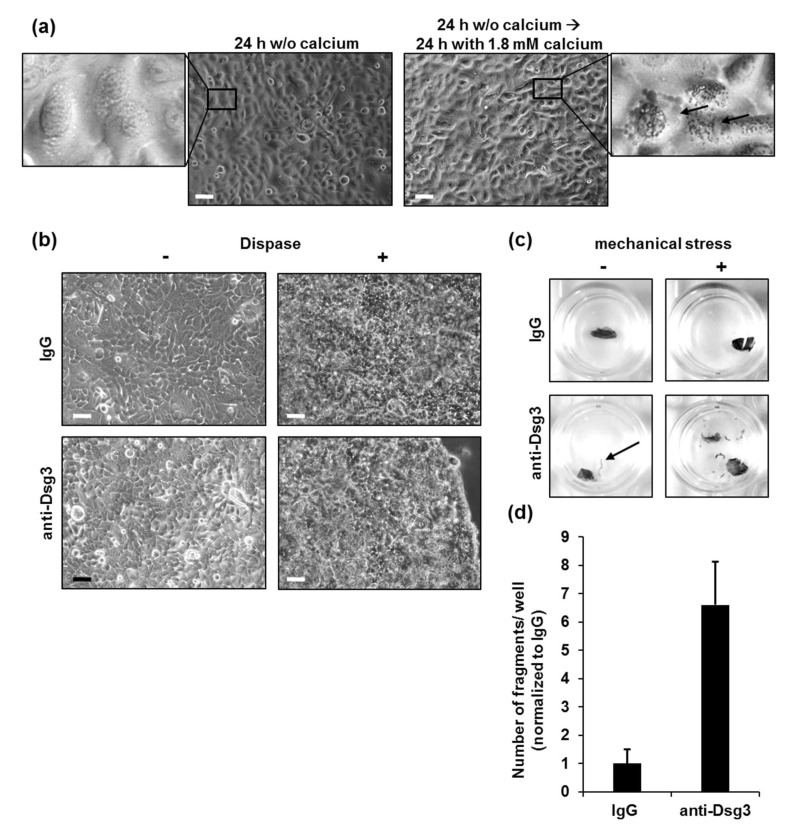
Optimal culturing conditions for generation of PK monolayers for DDA. (**a**) Human PK cultivated in calcium-free medium for 24 h (left panel) followed by cultivation in medium containing 1.8 mM of calcium for 24 h (right panel). Arrows in the enlarged image section indicate intracellular granules. The scale bars represent 50 µm. (**b**) Microscopic pictures of PK monolayers incubated with human IgG or anti-Dsg3 antibody before (-) and after (+) treatment with dispase. The scale bars represent 50 µm. (**c**) Macroscopic pictures of detached PK monolayers treated with IgG or anti-Dsg3 antibody, before (-) or after (+) application of mechanical stress. Arrow indicates an edge-originated fragment. (**d**) Quantification of the monolayer fragments. Anti-Dsg3 antibody-treated monolayers served as a pipetting control. One representative of three independent experiments is shown (triplicates within one single experiment were analyzed). Error bars represent the SEM.

**Figure 2 vaccines-10-00208-f002:**
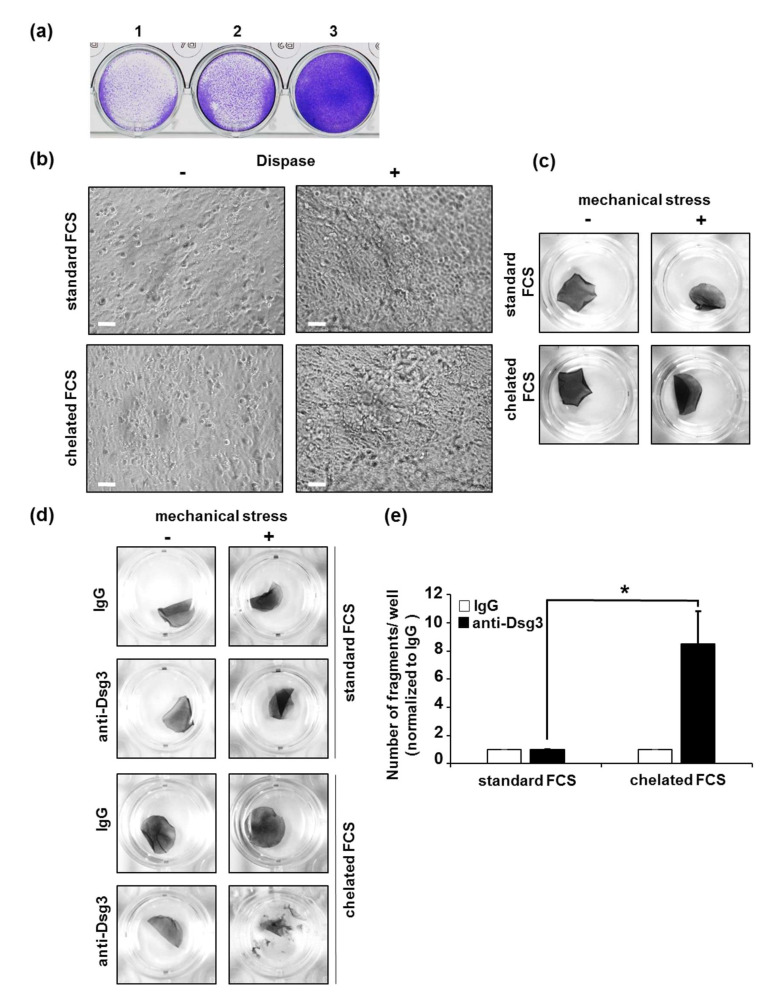
Optimal culturing conditions for generation of HaCaT monolayers for DDA. (**a**) Different amounts of HaCaT cells per well (1: 1.5 × 10^5^ cells/well; 2: 3 × 10^5^ cells/well; 3: 6 × 10^5^ cells/well), were cultivated overnight in medium containing 10% FCS and visualized by CV staining. (**b**) Microscopic pictures of HaCaT cells (initial number 6 × 10^5^) cultivated overnight as indicated, before (-) and after (+) treatment with dispase. The scale bars represent 50 µm. (**c**) Macroscopic pictures of detached HaCaT monolayers cultivated under indicated calcium conditions, before (-) or after (+) application of mechanical stress. (**d**) Macroscopic pictures of detached HaCaT monolayers cultivated under indicated calcium conditions and further stimulated with IgG or anti-Dsg3 antibody, before (-) and after (+) application of mechanical stress. (**e**) Quantification of the monolayer fragmentation in three independent experiments. Error bars represent the SEM. * *p* < 0.05.

**Figure 3 vaccines-10-00208-f003:**
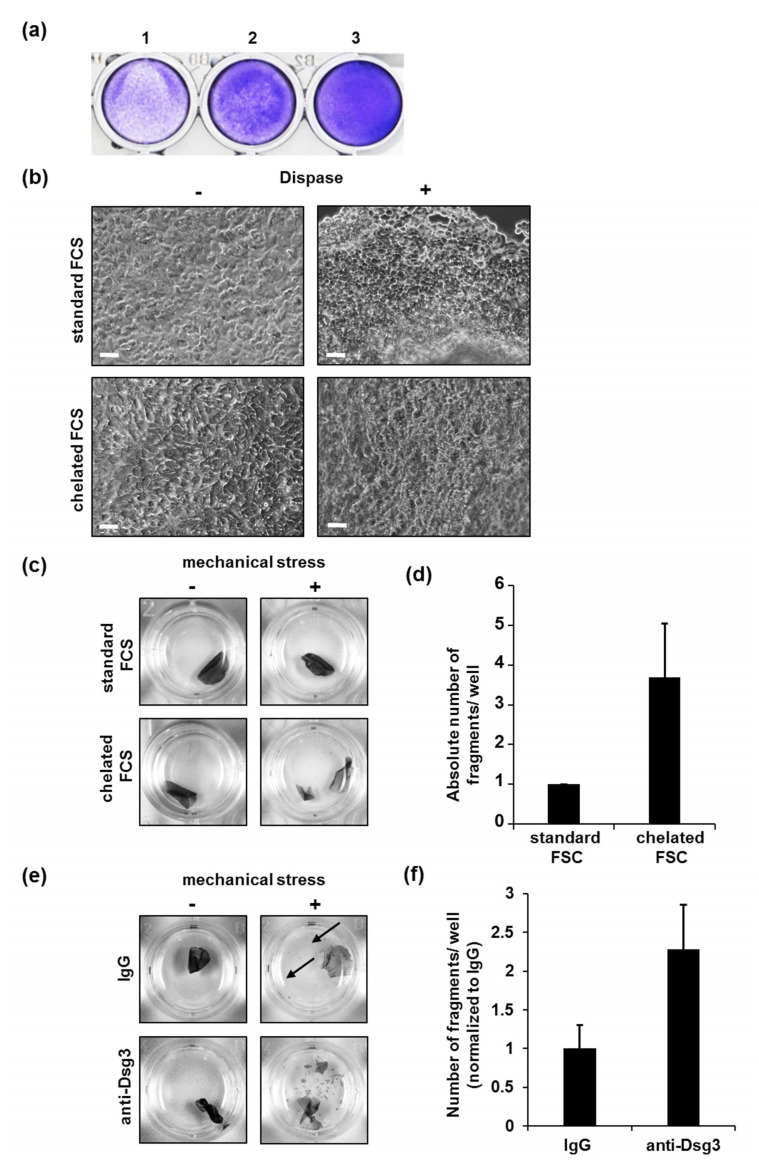
Optimal culturing conditions for generation of HaSKpw monolayers for DDA. (**a**) Different amounts of HaSKpw cells per well (1: 1.5 × 10^5^ cells/well; 2: 3 × 10^5^ cells/well; 3: 6 × 10^5^ cells/well) were cultivated overnight in medium containing 10% FCS and visualized by CV staining. (**b**) Microscopic pictures of HaSKpw cells (initial number 6 × 10^5^) cultivated for 48 h as indicated, before (-) and after (+) treatmentwith dispase. The scale bars represent 50 µm. (**c**) Macroscopic pictures of detached HaSKpw monolayers cultivated under indicated calcium conditions, before (-) or after (+) application of mechanical stress. (**d**) Quantification of the monolayer fragmentation in three independent experiments. Error bars represent the SEM. (**e**) Macroscopic pictures of detached HaSKpw monolayers cultivated as indicated and further stimulated with IgG or anti-Dsg3 antibody, before (-) or after (+) application of mechanical stress. Arrows indicate very small edge-originated fragments. (**f**) Quantification of the monolayer fragments. Anti-Dsg3 antibody-treated monolayers served as the pipetting control. One representative of three independent experiments is shown (duplicates within one single experiment were analyzed). Error bars represent the SEM.

**Figure 4 vaccines-10-00208-f004:**
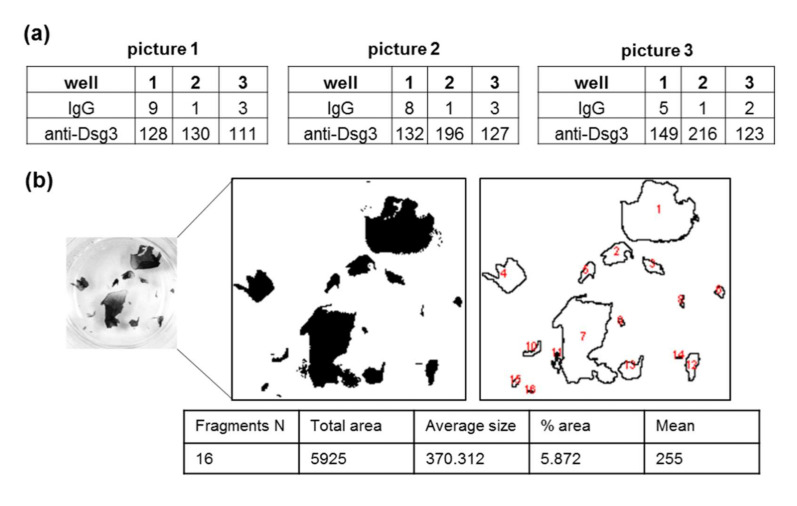
Quantification of fragments. (**a**) Number of fragments after counting of three independent pictures from the same experiment. (**b**) Sample of the fragments’ counting and quantification procedure using ImageJ software.

**Figure 5 vaccines-10-00208-f005:**
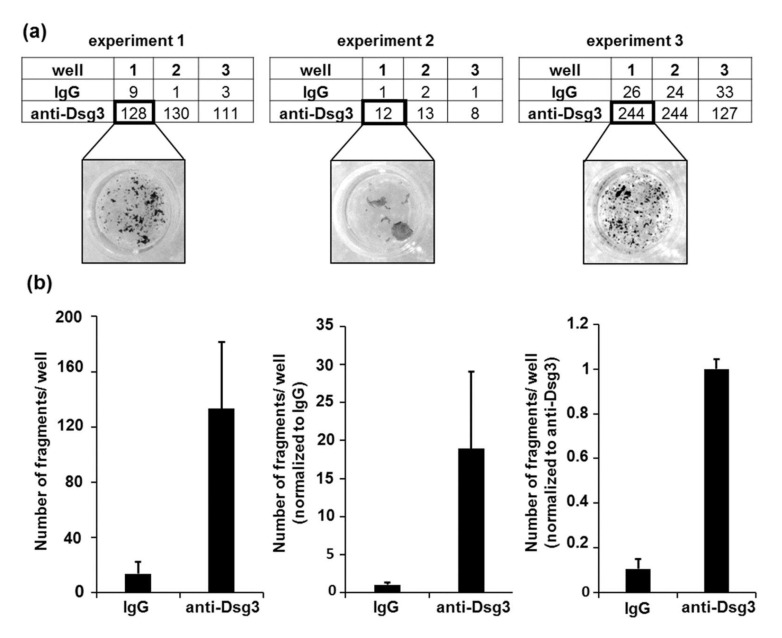
Data processing and visualization. (**a**) Number of fragments in three independent experiments. (**b**) Data evaluation and presentation via different normalization approaches. Error bars represent the SEM.

## Data Availability

Not applicable.

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
