# Peer review of "Pitfalls in the Application of Dispase-Based Keratinocyte Dissociation Assay for In Vitro Analysis of Pemphigus Vulgaris"

_vaccines, 2022, doi:10.3390/vaccines10020208_

Round 1
Reviewer 1 Report
Pitfalls in the application of dispase-based keratinocyte dissociation assay for in vitro analysis of pemphigus vulgaris
Morna F. Schmidt, Maria Feoktistova, Diana Panayotova-Dimitrova, Ramona A. Eichkorn and Amir S. Yazdi
Herewith I am submitting my reviewer comments for the above-mentioned manuscript, which is under consideration to be published in Vaccines. The article is about the dispase-based keratinocyte dissociation assay (DDA) and problems associated with it. While there is already an assay but the authors make an effort to systematically investigate different aspects of the assay and evaluate the outcome, which to my opinion is a good thing to do. Overall, I believe that this is an important topic with clinical relevance. The article is overall well written and clear. The level of English is sufficient (as far as I can tell). The figures are appealing and clear.
I only have a few very minor comments outlined below.
Line 54: “Dispase-based keratinocyte dissociation assay (DDA) is currently the main ” should be “The dispase-based keratinocyte dissociation assay (DDA) is currently the main ”
The materials and methods section is organized in an unusual way. I am not sure if the bullet points are in line with the journal format requirement. Please check.
Line 142: “Primary human keratinocytes were isolated and pooled from different donors.” Could you please comment on why the cells were pooled? Is there not enough material per donor? So is this always needed then? Or is it a way to have an average outcome?
I am wondering what this has to do with vaccines and if this paper is a good fit for this journal.
I have absolutely no complaints about the quality so this is more a question for the editors/authors to answer if this topic fits in the scope.
Author Response
Response to Reviewer 1 Comments
Herewith I am submitting my reviewer comments for the above-mentioned manuscript, which is under consideration to be published in Vaccines. The article is about the dispase-based keratinocyte dissociation assay (DDA) and problems associated with it. While there is already an assay but the authors make an effort to systematically investigate different aspects of the assay and evaluate the outcome, which to my opinion is a good thing to do. Overall, I believe that this is an important topic with clinical relevance. The article is overall well written and clear. The level of English is sufficient (as far as I can tell). The figures are appealing and clear.
I only have a few very minor comments outlined below.
Response: We would like to thank the reviewer for the constructive and valuable remarks and questions, and for the positive assessment of our manuscript. In the following, we will address the individual comments one by one.
- Line 54: “Dispase-based keratinocyte dissociation assay (DDA) is currently the main” should be “The dispase-based keratinocyte dissociation assay (DDA) is currently the main”.
Response 1: We have now corrected the sentence as suggested.
- The materials and methods section is organized in an unusual way. I am not sure if the bullet points are in line with the journal format requirement. Please check.
Response 2: We thank the reviewer for this comment. Our article aims to guide the reader/user through the applied method in an understandable and step-by-step manner, pointing out the pitfalls and optimal experimental conditions. For this purpose, we have adapted the structure of the section "Methods and Materials" in such a way that, on the one hand, the experimental setup as well as the execution of the experiments and, on the other hand, the detailed step-by-step implementation of the method become comprehensible. To illustrate this, basic features of a protocol are implemented in this section.
- Line 142: “Primary human keratinocytes were isolated and pooled from different donors.” Could you please comment on why the cells were pooled? Is there not enough material per donor? So is this always needed then? Or is it a way to have an average outcome?
Response 3: We thank the reviewer for this point. The aim of pooling keratinocytes from different donors is to avoid experimental inconsistencies due to individual factors/characteristics of individual donors. Furthermore, the aim is to achieve highly reproducible data corresponding to the cell type in general. We have now included a detailed explanation in section “Materials and Methods” as follows:
“Primary human keratinocytes were isolated and pooled from different donors to minimize individual-specific influencing factors and to achieve an average outcome with reproducible experimental conditions.”
- I am wondering what this has to do with vaccines and if this paper is a good fit for this journal.
I have absolutely no complaints about the quality so this is more a question for the editors/authors to answer if this topic fits in the scope.
Response 4: We thank the reviewer for this remark. The article is submitted to be included in the special issue on "Biological/Targeted Therapy of Immune-Mediated Skin Diseases". Therefore, a method to experimentally study autoimmune skin diseases might be of research interest. Suitable tools to investigate the complex pathophysiological relationships are indispensable as an experimental platform. For this very reason, we are convinced that our article/protocol fits very well into an overview of current research in this field and provides a suitable contribution.
Reviewer 2 Report
The manuscript by Schmidt et al. provides protocols and describes standardized procedures for different keratinocytes and cell lines to perform dispase-based dissociation assays. This technique is a common approach in the field of cell-cell adhesion and indeed would benefit from standardization over different labs. As such, I feel that a more thorough analysis of the different variables (Ca2+ concentrations, cultivation time, settings for automated image analysis etc.) would be beneficial to provide such standardization and identify biases. Still, the manuscript is helpful and provides a good starting reference for persons doing this type of assay for the first time.
Specific comments:
- A major confounder in dispase-based dissociation assays is the serial and manual application of stress (either using a mechanical or an electrical pipette). Thus, a technique prone to less bias, by which all wells receive a similar amount of stress simultaneously, would be preferred. The authors should include additional experiments using orbital shakers or rotators and compare these to the standard approach.
- The authors state that “pseudo-fragments” after shearing should be excluded. How do the authors determine that these originate from cell borders only? And even if so, why is this a reason for exclusion? Apparantly, the resilience to shear stress is not fully uniform throughout a monolayer and including all fragments detectable at a given magnification should not necessarily impair the interpretability of the results.
- How do the authors determine “confluency”, as this is to some extent a continuum? Criteria outlining when a monolayer is “fit” for a dissociation assay would help the reader to determine a good time point for stimulation.
- Error bars are not always visible (e.g. 1d, 3f) Please plot data with single data points.
- Scale bars are not readable and in part missing in the brightfield images.
- 7: “[…] as desmosomal adhesive strength is calcium-dependent.” This is only partially true, as desmosomes can be calcium-independent (“hyperadhesive”). A better statement may be “[…] as desmosome formation is calcium-dependent.”
Author Response
Response to Reviewer 2 Comments
The manuscript by Schmidt et al. provides protocols and describes standardized procedures for different keratinocytes and cell lines to perform dispase-based dissociation assays. This technique is a common approach in the field of cell-cell adhesion and indeed would benefit from standardization over different labs. As such, I feel that a more thorough analysis of the different variables (Ca2+ concentrations, cultivation time, settings for automated image analysis etc.) would be beneficial to provide such standardization and identify biases. Still, the manuscript is helpful and provides a good starting reference for persons doing this type of assay for the first time.
Response: We would like to thank the reviewer for the constructive and valuable remarks, and the positive assessment of our manuscript. We agree with the reviewer that thorough analysis of different variables such as Ca2+ concentrations, cultivation time and settings for automated image analysis would be further helpful and beneficial to provide such standardization and identify biases. However, the scope of this manuscript was not to investigate a broader spectrum of new conditions, but rather to focus on, optimize and combine the most often applied conditions and settings for DDA. Thus, we think that our protocol can provide a higher inter-experimental comparability not only for persons applying the assay for the first time, but especially in the routine application of the method, even in experienced laboratories. Therefore, we believe that the main disadvantage of DDA, namely the high variability of different operating conditions, such as the stability of the monolayer, could be overcome even when reproduced in the same laboratory.
- A major confounder in dispase-based dissociation assays is the serial and manual application of stress (either using a mechanical or an electrical pipette). Thus, a technique prone to less bias, by which all wells receive a similar amount of stress simultaneously, would be preferred. The authors should include additional experiments using orbital shakers or rotators and compare these to the standard approach.
Response 1: Since we aimed to use and optimize most used experimental settings, we choose to concentrate on the use of an electrical pipette for the application of mechanical stress. We completely agree with the reviewer that information regarding this method to apply mechanical stress should be included in our manuscript. We have now added the following sentences in the “Results” part of the manuscript:
“Based on existing approaches, we have focused on the application of mechanical stress using an electrical pipette. Alternative approaches for application of mechanical stress via rotator/shaker were previously published [16 (PMID: 12499357), 29 (PMID: 15304078), 30 (PMID: 21156808)]. However, they include transfer steps of the cell monolayer, resulting in an additional stress which cannot be standardized between all analyzed conditions. The ultimate advantage of using an electrical pipette is the possibility to predefine exact settings, such as speed or volume and to apply the very same stress level to all analyzed conditions.”
- The authors state that “pseudo-fragments” after shearing should be excluded. How do the authors determine that these originate from cell borders only? And even if so, why is this a reason for exclusion? Apparantly, the resilience to shear stress is not fully uniform throughout a monolayer and including all fragments detectable at a given magnification should not necessarily impair the interpretability of the results.
Response 2: We thank the reviewer for this very important point. The “pseudo-fragments” detected in PK monolayers were visible before the application of mechanical stress. Therefore, we have acknowledged these fragments as non-specific and not resulting from the applied mechanical stress. Consequently, we have excluded them from the quantification. In the case of HaSKpw cell line, we observed the “pseudo-fragments” after the application of mechanical stress, but in the IgG-stimulated control condition. Since the monolayer was still intact and stable upon mechanical stress application, and the size of these particles was very small and incomparable to usual appearance of layer fragments, we acknowledged these small particles as edge-originated in terms of layer-fragments shedding from the boarders of an intact and stable monolayer. Therefore, these non-specific fragments were excluded from quantification in all analyzed conditions.
We excuse for the not clear enough description in the submitted manuscript. We have now included more detailed explanations regarding the exclusion of the non-specific fragment in “Results” part 3.4 as follows:
“A critical step during quantification of the monolayer fragments is considering the impact of the cell specific characteristics of the respective cell line which can influence the qualities of the monolayer. Our experiments with PK demonstrated very rarely the appearance of single fragments prior to application of mechanical stress (Figure 1c, panel left, down). We considered these fragments as non-specific “pseudo-fragments” which originated from the edges of an intact monolayer and excluded them from the following quantification. Additionally, we demonstrated that monolayers generated from HaSKpw cells were less stable and more fragile in comparison to PK or HaCaT cells. This is due to individual cell-specific properties of HaSKpw cells as discussed below. Our experiments demonstrated that application of mechanical stress in control conditions (IgG stimulation) also resulted in very small fragments (Figure 3e). Since these small particles were detected in the presence of an otherwise intact monolayer, we hypothesized that they do originate from the edges of the monolayer. Therefore, we have acknowledged them as non-specific “pseudo-fragments”. ImageJ software settings has been adjusted so that all fragments smaller than the largest unspecific fragments in control conditions performed with HaSKpw were excluded and applied to all analyzed conditions.”
- How do the authors determine “confluency”, as this is to some extent a continuum? Criteria outlining when a monolayer is “fit” for a dissociation assay would help the reader to determine a good time point for stimulation.
Response 3: We thank the reviewer for this very important suggestion. Now we have included in “Results” parts 3.1, 3.2 and 3.3 additional sentences explaining more precisely the difference between “confluent” and “fit” for DDA cellular layers.
- Error bars are not always visible (e.g. 1d, 3f) Please plot data with single data points.
Response 4: We thank the reviewer for this point. We have now improved figures 1d and 3f. However, since we show one of three experiments, we felt that this data representation is more appropriate.
- Scale bars are not readable and in part missing in the brightfield images.
Response 5: We have now improved the scale bars.
- “[…] as desmosomal adhesive strength is calcium-dependent.” This is only partially true, as desmosomes can be calcium-independent (“hyperadhesive”). A better statement may be “[…] as desmosome formation is calcium-dependent.”
Response 6: We have improved the sentence as suggested by the reviewer.
Round 2
Reviewer 2 Report
The authors did not, as suggested, include additional experiments to evaluate other, more standardized ways to apply stress to monolayers. The additional sentences with regard to this topic provided by the authors are not satisfactory, as they suggest that using an electrical pipette is the gold-standard (it is better than a mechanical pipette, but still providing considerable inter-well variation, e.g., by holding the pipette in a different angle or approaching the monolayer at different positions). Moreover, using plates on an orbital shaker does not, as indicated by the authors, require transfer of fragments. Please adapt.
The introduced sentences on the rationale for exclusion of fragments does not, in the case of HaSKpw, make it more convincing and might be misleading. As stress is applied to an intact monolayer, resulting fragments are certainly not “non-specific”, but rather a result of adhesive heterogeneity. I think what the authors mean, and where they may be right, is that it is a difference in overall adhesion if you have a monolayer with 1 large and 9 very small fragments vs. a monolayer with 10 medium-size fragments. In such cases, it might be best to apply a size-cutoff to all wells, as suggested by the authors, or better to show fragment numbers as histogram with respect to fragment size. Still, the most unbiased approach would be to (i) exclude wells in which small fragments appear after dispase digestion and before application of shear stress (monolayers do not reach the quality criterium), (ii) count ALL fragments detectable after application of shear stress and (iii) apply a statistically meaningful amount of technical and biological replicates to account for monolayer heterogeneity and other confounders.
As the authors intend this manuscript as a guide to others, this “pitfall” should be discussed more thoroughly.
